analytical chemistry

bioactivation, LC–MS/MS, epidermal growth factor receptor, human liver microsomes, iminium ion intermediates, lung cancer

**Author for correspondence:**
Mohamed W. Attwa
e-mail: mzeidan@ksu.edu.sa

This article has been edited by the Royal Society of Chemistry, including the commissioning, peer review process and editorial aspects up to the point of acceptance.

# Liquid chromatography–tandem mass spectrometry metabolic profiling of nazartinib reveals the formation of unexpected reactive metabolites

Ali S. Abdelhameed[1], Mohamed W. Attwa[1,2]
and Adnan A. Kadi[1]

[1]Department of Pharmaceutical Chemistry, College of Pharmacy, King Saud University, PO Box 2457, Riyadh 11451, Kingdom of Saudi Arabia
[2]Students' University Hospital, Mansoura University, Mansoura 35516, Egypt

MWA, 0000-0002-1147-4960

Nazartinib (EGF816, NZB) is a promising third-generation human epidermal growth factor receptor (EGFR) tyrosine kinase inhibitor. This novel irreversible mutant-selective EGFR inhibitor targets EGFR containing both the resistance mutation (T790M) and the activating mutations (L858R and Del19), while it does not affect wild-type EGFR. However, the metabolic pathway and bioactivation mechanisms of NZB are still unexplored. Thus, using liquid chromatography–tandem mass spectrometry, we screened for products of NZB metabolism formed *in vitro* by human liver microsomal preparations and investigated the formation of reactive intermediates using potassium cyanide as a nucleophile trap. Unexpectedly, the azepane ring was not bioactivated. Instead, the carbon atom between the aliphatic linear tertiary amine and electron-withdrawing system (butenoyl amide group) was bioactivated, generating iminium intermediates as reactive species. Six NZB phase I metabolites, formed by hydroxylation, oxidation and *N*-demethylation, were characterized. Moreover, two reactive iminium ions were characterized and their corresponding bioactivation mechanisms were proposed. Based on our results, we speculate that bioactivation of NZB can be blocked by small sterically hindering groups, isosteric replacement or a spacer. This approach might reduce the toxicity of NZB by avoiding the generation of reactive species.

# 1. Introduction

Non-small-cell lung cancer (NSCLC) encompasses a heterogeneous group of lung cancer subtypes [1–5], which affects 90% of patients with lung cancer [6]. This class of lung cancer is associated with several mutations, such as those in human epidermal growth factor receptor (EGFR). Tyrosine kinase inhibitors (TKIs) regulate the activity of human EGFR and have become the standard treatment for patients suffering from advanced EGFR-mutant NSCLC. The first-generation EGFR TKIs (e.g. gefitinib and erlotinib) bind reversibly and competitively to the ATP-binding site of the EGFR tyrosine kinase (TK) domain, which improves the outcome of NSCLC patients bearing EGFR-activating mutations (L858R and Del19) [7,8]. However, after satisfactory responses for a period, patients' tumours acquired resistance to first-generation TKIs because of the development of a T790M mutation, which affects the ATP-binding site of the human EGFR [9–12].

Thus, second-generation EGFR TKIs (e.g. avitinib and dacomitinib) were designed to target tumours with T790M mutation and EGFR-activating mutations. These compounds showed promising anti-T790M activity in laboratory experiments. However, their clinical activity towards T790M-associated NSCLC was limited because of their inhibitory effects on wild-type EGFR, which resulted in toxicity and a narrow therapeutic index [13–15]. More recently, third-generation EGFR TKIs (e.g. osimertinib and nazartinib (NZB)) were developed. They irreversibly and selectively target EGFR with T790M and other mutations, whereas they have little effect on wild-type EGFR activity [13,14]. Third-generation EGFR TKIs were developed to overcome EGFR T790M-mediated resistance to first- and second-generation EGFR TKIs with minor toxicity. Third-generation EGFR TKIs combine effectiveness against NSCLC that is resistant to both first- and second-generation EGFR TKIs [16,17]. Osimertinib, for example, is approved by both the American and European regulatory agencies for the management of patients with metastatic EGFR T790M NSCLC [18]. Pre-clinical data show that NZB, another third-generation EGFR TKI [19], does not affect wild-type EGFR activity and presents selectivity against mutated EGFR, similar to other third-generation EGFR TKIs. Nevertheless, it presents some side effects, such as diarrhoea, pruritus and rash [20].

In addition to the drug itself, by-products of detoxification pathways may be responsible for such adverse effects in patients. Detoxification involves metabolic reactions that transform endogenous compounds and xenobiotics, increasing their polarity to be excreted from the human body. Although metabolites usually exhibit less toxicity than their parents, in some cases, bioactivation may generate reactive intermediates that are more toxic than the unmetabolized molecules [21–23]. Reactive intermediates are unstable and can modify DNA and proteins by the formation of covalent bonds, which is considered the initial step in drug-induced organ toxicity [24,25]. Thus, the identification of generated reactive metabolites is crucial for understanding drug-induced toxicity. However, reactive metabolites are usually generated by phase I metabolic pathways and their identification is hindered by their transient nature. To overcome this limitation, a nucleophile can be used to capture reactive intermediates, and the resulting adducts can be characterized and identified by mass spectrometry technique [26,27].

The chemical structure of NZB (N-(7-chloro-1-{(3R)-1-[(2E)-4-(dimethylamino)-2-butenoyl]-3-azepanyl}-1H-benzimidazole-2-yl)-2-methyl isonicotinamide; figure 1) contains two tertiary nitrogen atoms (an azepane ring and a terminal dimethylamino group) that can be bioactivated, generating iminium ion intermediates [28–31]. The formation of unstable intermediates reveals side effects of NZB as was approved with similar drugs. Cyclic tertiary amine rings can perform bioactivation by iminium ion generation [28–31]. These intermediates react poorly with glutathione; however, they can be trapped using potassium cyanide [21,28,29]. The obtained reactive iminium intermediates trapped efficiently using cyanide to form cyano conjugates can be characterized by mass spectrometry [26–28,32,33]. Moreover, although the azepane ring was expected to undergo bioactivation during NZB metabolism, this does not occur. Instead, the carbon between the aliphatic linear tertiary amine and the unsaturated conjugated system are bioactivated.

It is hypothesized that these reactive metabolites might be responsible for the side effects of NZB. However, there are no reports on specific metabolic pathways associated with the bioactivation mechanism of NZB. Thus, the aim of this work was to use *in vitro* experiments to characterize the bioactivation pathways of NZB that form reactive intermediates. To do so, we used a scavenging molecule (potassium cyanide) to trap reactive intermediates of NZB metabolism. This approach was used because when reactive metabolites form *in vivo*, they bind to DNA and proteins via covalent bonds and hence cannot be detected [24,27,32].

**Figure 1.** Chemical structure of NZB showing its building blocks.

# 2. Material and methods

## 2.1. Chemicals

NZB was obtained from MedChem Express (Monmouth Junction, NJ, USA). Formic acid, ammonium formate, potassium cyanide, pooled human liver microsomes (HLMs, M0567) and acetonitrile were procured from Sigma-Aldrich (St Louis, MO, USA). High-performance liquid chromatography (HPLC)-grade water ($H_2O$) was generated by an in-house Milli-Q Plus purification system (Burlington, MA, USA). All other solvents and chemicals were of analytical grade.

## 2.2. Chromatographic conditions

Resolution and identification of *in vitro* NZB metabolites and its related cyano adducts from the HLM incubation mixtures was performed on an Agilent Triple Quadrupole system comprising an Agilent rapid resolution liquid chromatography (RRLC) 1200 as an HPLC system and an Agilent 6410 triple quadrupole (QqQ) as a mass detector (Agilent Technologies, Palo Alto, CA, USA) with an electrospray ionization (ESI) source. Chromatographic resolution of the metabolic mixtures components was done on a $C_{18}$ column (length, 150 mm; internal diameter, 2.1 mm; and particle size, 3.5 µm). The column temperature was fixed at $22 \pm 1°C$, and we used a gradient mobile phase at a flow rate of $0.2$ ml min$^{-1}$ and consisting of 10 mM ammonium formate (solvent A; pH 4.2) and acetonitrile (solvent B). The gradients steps involved solvent B (5%; 0–5 min), solvent B (5–50%; 5–35 min), solvent B (50–90%; 35–50 min) and solvent B (90–5%; 50–60 min), with a post time of 15 min. The sample injection volume was 10 µl. The run time was 60 min, with the chromatographic and mass parameters preoptimized for NZB. The generation of daughter ions (DIs) of NZB metabolites and cyano adducts was done in the collision cell by collision-induced dissociation (CID). Mass analysis was performed on a mass detector using positive ESI source. Nitrogen ($N_2$) was used as drying gas at a flow rate of 11 l min$^{-1}$, and as collision gas at a pressure of 55 psi. Capillary voltage, source temperature, fragmentor voltage and collision energy were set to 4000 V, 350°C, 140 V and 18 eV, respectively. Agilent Mass Hunter software was used for controlling instrument and data acquisition.

## 2.3. Human liver microsomes incubation

We first exposed HLMs to several NZB concentrations (2–30 µM) and found that the composition of metabolites did not vary within this range. However, the concentration of metabolites increased as the concentration of NZB increased. Thus, 30 µM was used in all experiments to increase the yield of metabolites and make their characterization easier. The screening of NZB metabolites was performed *in vitro* by incubating NZB (30 µM) with HLMs (1.0 mg ml$^{-1}$) in phosphate buffer (50 mM at pH 7.4) and $MgCl_2$ (3.3 mM) for 120 min at 37°C in a shaking water bath. The *in vitro* metabolization of NZB was stimulated by the addition of NADPH (1.0 mM) and terminated by the addition of ice-cold acetonitrile [34,35]. The same HLM incubation experiment was repeated in the presence of potassium cyanide to capture the reactive intermediates. All reactions were performed in triplicate to verify the

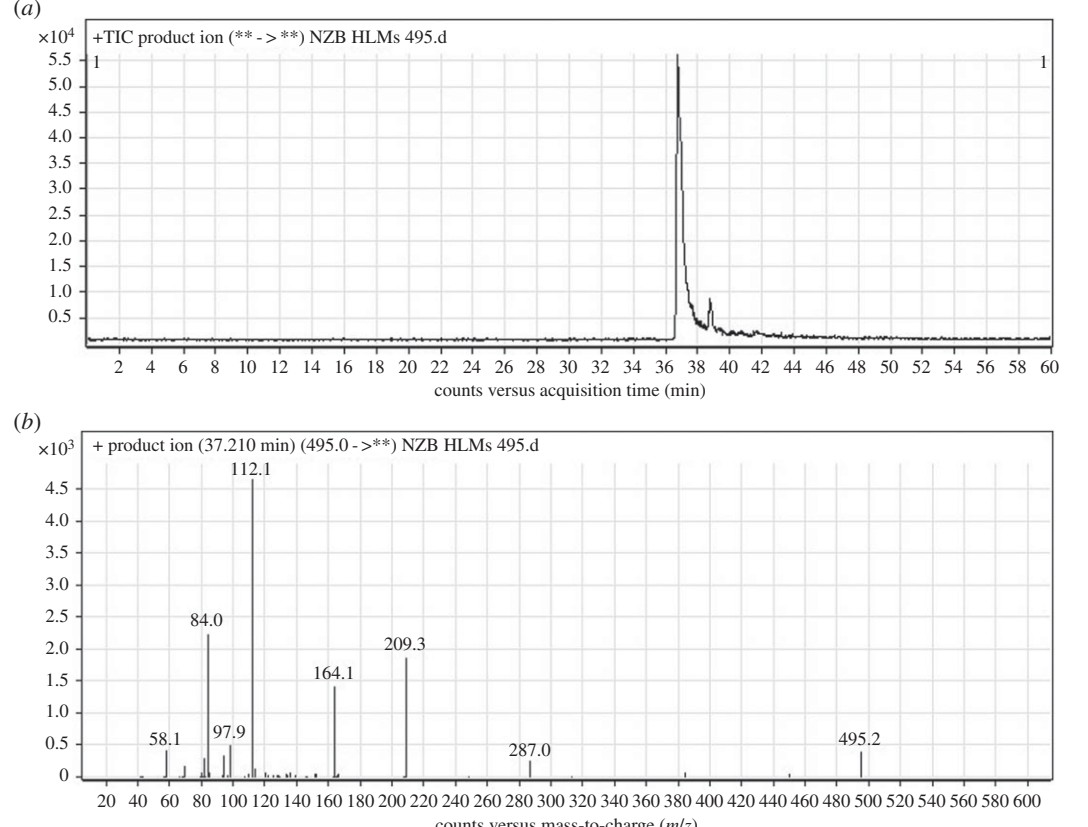

**Figure 2.** PI chromatogram of NZB (*a*) and DIs mass spectrum at 37.21 min (*b*).

results. The purification of the incubated solutions was performed by a protein precipitation method involving: (i) centrifugation at 9000*g* for 15 min at 4°C, (ii) the transfer of the supernatant into clean vials, and (iii) concentration of the extracts by evaporation under nitrogen stream, followed by reconstitution in 0.5 ml of mobile phase. To analyse the composition of each sample, 10 µl was injected into a liquid chromatography tandem mass spectrometry (LC–MS/MS) [36–38]. Controls were prepared following the same steps except the addition of the drug or NADPH.

## 2.4. Identification of NZB reactive intermediates

Full mass spectrometry scans and extracted ion chromatograms of the detected mass to charge ratio (*m/z*) peaks were used to identify the *in vitro* metabolites in the incubation mixtures. Molecular ions were used as parent ions (PIs) for fragmentation into daughter ions (DIs). The fragmentation behaviour was used to characterize the reactive metabolites formed during NZB metabolism by HLMs *in vitro*.

# 3. Results and discussion

## 3.1. Fragmentation analysis of NZB

The chemical structure of NZB contains five building blocks (isonicotinamide, benzimidazole, azepane, tertiary dimethyl amine and butenoyl). The fragmentation of the NZB PI generated qualitative DIs that were used to identify the metabolic changes in the NZB structure. The NZB PI peak eluted at 37.21 min (figure 2*a*). The fragmentation of the PI at *m/z* 495 generated six DIs at *m/z* 287, *m/z* 209, *m/z* 164, *m/z* 112, *m/z* 84 and *m/z* 58 (figure 2*b*). The DI at *m/z* 287 was used to trace any changes on the isonicotinamide and benzimidazole groups. The DIs at *m/z* 209 and *m/z* 164 were used to trace any changes on the azepane ring. The DIs at *m/z* 112 and *m/z* 84 were used to trace any changes on the butenoyl group. The DI at *m/z* 58 was used to trace any changes on the dimethyl amine group (scheme 1).

**Scheme 1.** Fragmentation behaviour of NZB. Asterisk stands for reactive centre. DIs, daughter ions.

**Table 1.** *In vitro* phase I and reactive metabolites of NZB. MS, mass spectrometry; NZB, nazartinib; RT, retention time.

| molecule | MS scan | most abundant fragment ions (*m/z*) | RT (min) | metabolic reaction |
|---|---|---|---|---|
| *original drug* | | | | |
| NZB | 495 | 287, 209, 164, 112, 84, 58 | 37.21 | no reaction |
| *phase I metabolites* | | | | |
| NZB481 | 481 | 287, 195, 98, 44 | 36.12 | *N*-demethylation |
| NZB509a | 509 | 301, 209, 112, 84 | 30.19 | oxidation at the methyl attached to the isonicotinamide group |
| NZB509b | 509 | 287, 233, 126 | 36.72 | α-oxidation of the dimethyl amine group |
| NZB509c | 509 | 450, 353, 164, 120, 58 | 43.49 | α-oxidation at the azepane ring |
| NZB511a | 511 | 287, 225, 180, 112 | 32.76 | α-hydroxylation at the azepane ring |
| NZB511b | 511 | 303, 209, 112, 84 | 34.14 | hydroxylation at the methyl attached to the isonicotinamide group |
| *reactive metabolites* | | | | |
| NZB520 | 520 | 493, 207, 164, 83, 57 | 47.66 | cyano addition at the bioactivated carbon |
| NZB506 | 506 | 287, 220, 120, 98 | 48.95 | *N*-demethylation and cyano addition at the bioactivated carbon |

## 3.2. Characterization of phase I nazartinib metabolites and reactive intermediates

Phase I metabolic reactions (hydroxylation, oxidation and *N*-demethylation) produced six metabolites. In addition, we detected two reactive intermediates as cyano adducts (table 1).

### 3.2.1. Identification of the NZB481 phase I metabolite

The NZB481 PI peak eluted at 36.12 min (figure 3*a*). The fragmentation of the PI at *m/z* 481 generated four DIs at *m/z* 287, *m/z* 195, *m/z* 98 and *m/z* 44 (figure 3*b*). In comparison with the NZB fragmentation pattern, the DI at *m/z* 287 revealed no metabolic change on the isonicotinamide and benzimidazole groups. The DIs at *m/z* 195, *m/z* 98 and *m/z* 44 exhibited decreases of 14 *m/z* units. Thus, the DIs at *m/z* 98 and *m/z* 44 indicated that an *N*-demethylation metabolic change occurred on the dimethyl amine group (scheme 2).

### 3.2.2. Identification of the NZB509a and NZB509b phase I metabolites

The PI peaks of NZB509a, NZB509b and NZB509c eluted at 30.19, 36.72 and 43.49 min, respectively (figure 4*a*). The fragmentation of the PI at *m/z* 509 generated several DIs (figure 4*b–d*).

The fragmentation of NZB509a resulted in four DIs at *m/z* 301, *m/z* 209, *m/z* 112 and *m/z* 84 (figure 4*b*). In comparison with the NZB fragmentation pattern, the DIs at *m/z* 209, *m/z* 112 and *m/z* 84 revealed no metabolic change on the azepane ring, dimethyl tertiary amine group and butenoyl group. The DI at *m/z* 301 showed an increase of 14 *m/z* units, indicating that the methyl attached to the isonicotinamide group was oxidized during metabolism (scheme 3).

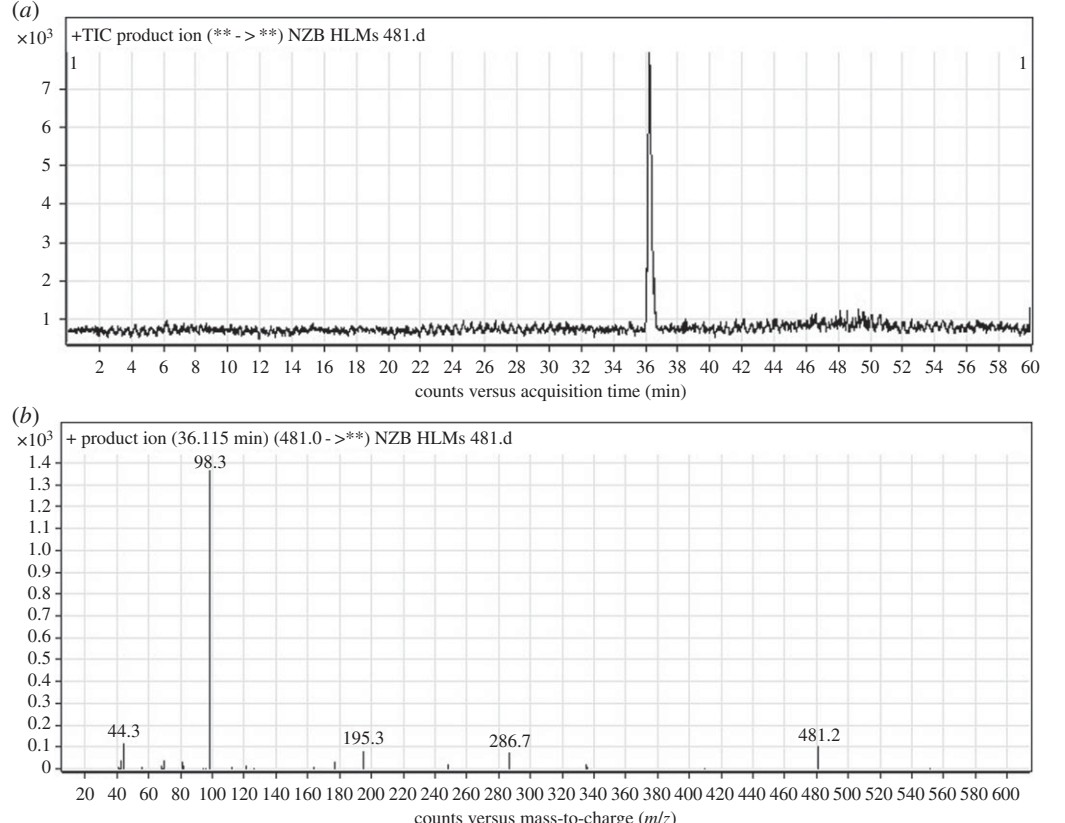

**Figure 3.** DI chromatogram of NZB481 (*a*) and DI mass spectrum at 36.12 min (*b*).

**Scheme 2.** Fragmentation behaviour of NZB481. DIs, daughter ions.

The fragmentation of NZB509b resulted in three DIs at *m/z* 287, *m/z* 223 and *m/z* 126 (figure 4*c*). In comparison with the NZB fragmentation pattern, the DI at *m/z* 287 indicated the absence of any metabolic change on the isonicotinamide and benzimidazole groups. The DIs at *m/z* 223 and *m/z* 126 showed increases of 14 *m/z* units. Thus, the DI at *m/z* 126 indicated that an oxidation metabolic reaction occurred on the carbon α of the dimethyl amine group (scheme 4).

The fragmentation of NZB509c resulted in five DIs at *m/z* 450, *m/z* 353, *m/z* 164, *m/z* 120 and *m/z* 58 (figure 4*d*). In comparison with the NZB fragmentation pattern, the DI at *m/z* 58 indicated that no metabolic change occurred on the dimethyl amine group. The DIs at *m/z* 450 and *m/z* 353 (resulting from a retro-Diels–Alder reaction) revealed the oxidation of the azepane ring, in agreement with the other DIs at *m/z* 120 and *m/z* 58 (scheme 5).

### 3.2.3. Identification of the NZB511a and NZB511b phase I metabolites

The NZB511a and NZB511b PI peaks appeared at 32.76 and 34.14 min, respectively (figure 5*a*). The fragmentation of the PI at *m/z* 511 produced various DIs (figure 5*b,c*).

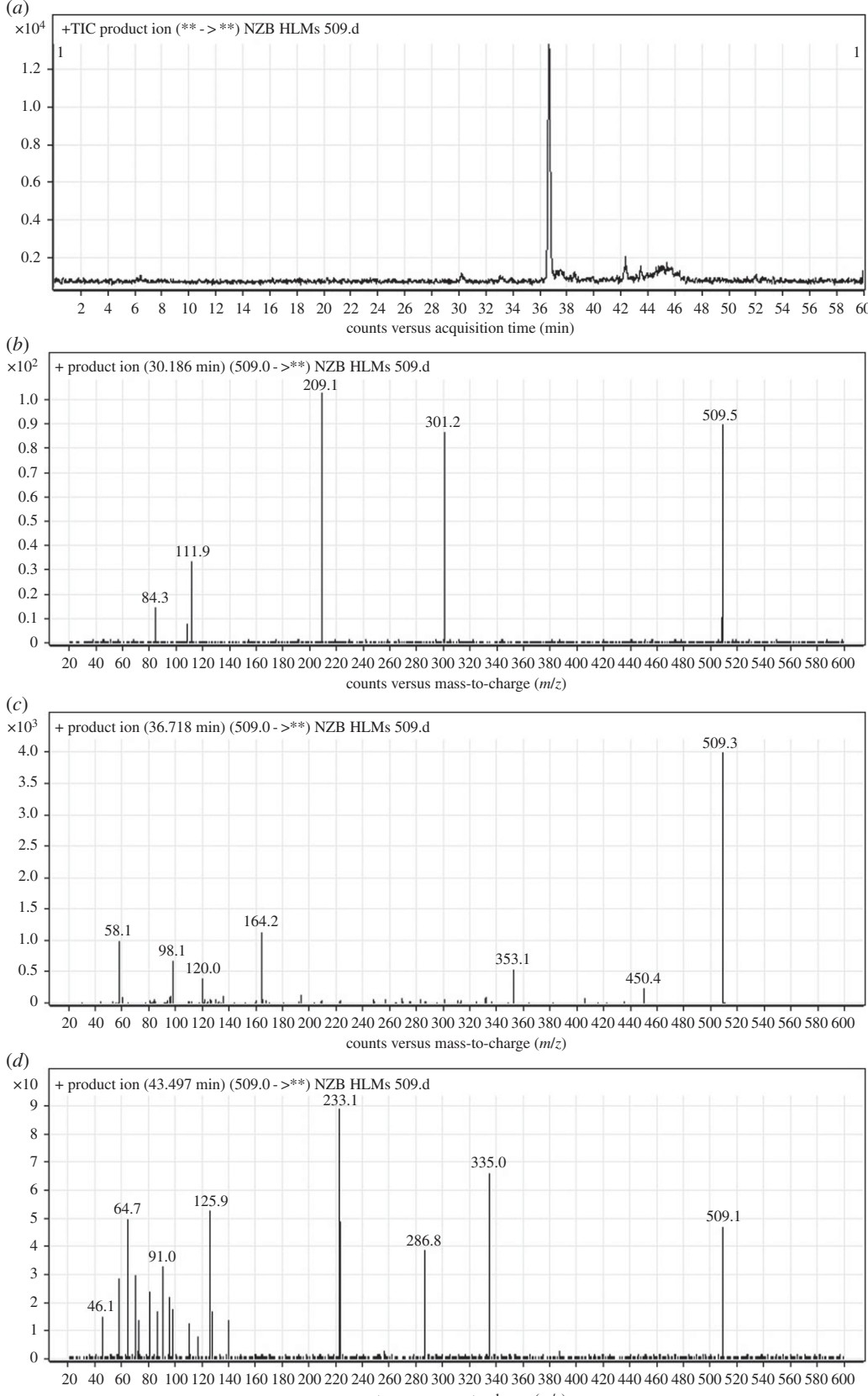

**Figure 4.** DI chromatogram of NZB509 metabolites (*a*) and DI mass spectra at 30.19 min (*b*), 36.72 min (*c*) and 43.49 min (*d*).

The fragmentation of NZB511a at *m/z* 511 resulted in four DIs at *m/z* 287, *m/z* 225, *m/z* 180 and *m/z* 112 (figure 5*b*). In comparison with the NZB fragmentation pattern, the DI at *m/z* 287 revealed the absence of any metabolic reaction at the isonicotinamide and benzimidazole groups, and the DI at *m/z* 112 indicated

**Scheme 3.** Fragmentation behaviour of NZB509a. DIs, daughter ions.

**Scheme 4.** Fragmentation behaviour of NZB509b. DIs, daughter ions.

**Scheme 5.** Fragmentation behaviour of NZB509c. DIs, daughter ions.

the absence of any metabolic reaction on the butenoyl group. The DIs at $m/z$ 225 and $m/z$ 180 showed increases of 16 $m/z$ units, indicating that hydroxylation occurred on the azepane ring (scheme 6).

The fragmentation of NZB511b resulted in four DIs at $m/z$ 303, $m/z$ 209, $m/z$ 112 and $m/z$ 84 (figure 5c). In comparison with the NZB fragmentation pattern, the DIs at $m/z$ 209, $m/z$ 112 and $m/z$ 84 indicated the absence of any metabolic reaction on the azepane ring, the dimethyl amine group and the butenoyl group. The DI at $m/z$ 303 showed an increase of 16 $m/z$ units, indicating hydroxylation on the methyl attached to the isonicotinamide group (scheme 7).

## 3.3. Reactive metabolites

In addition to the metabolites described above, two cyano adducts were characterized, indicating the generation of reactive intermediates in NZB metabolism by HLMs.

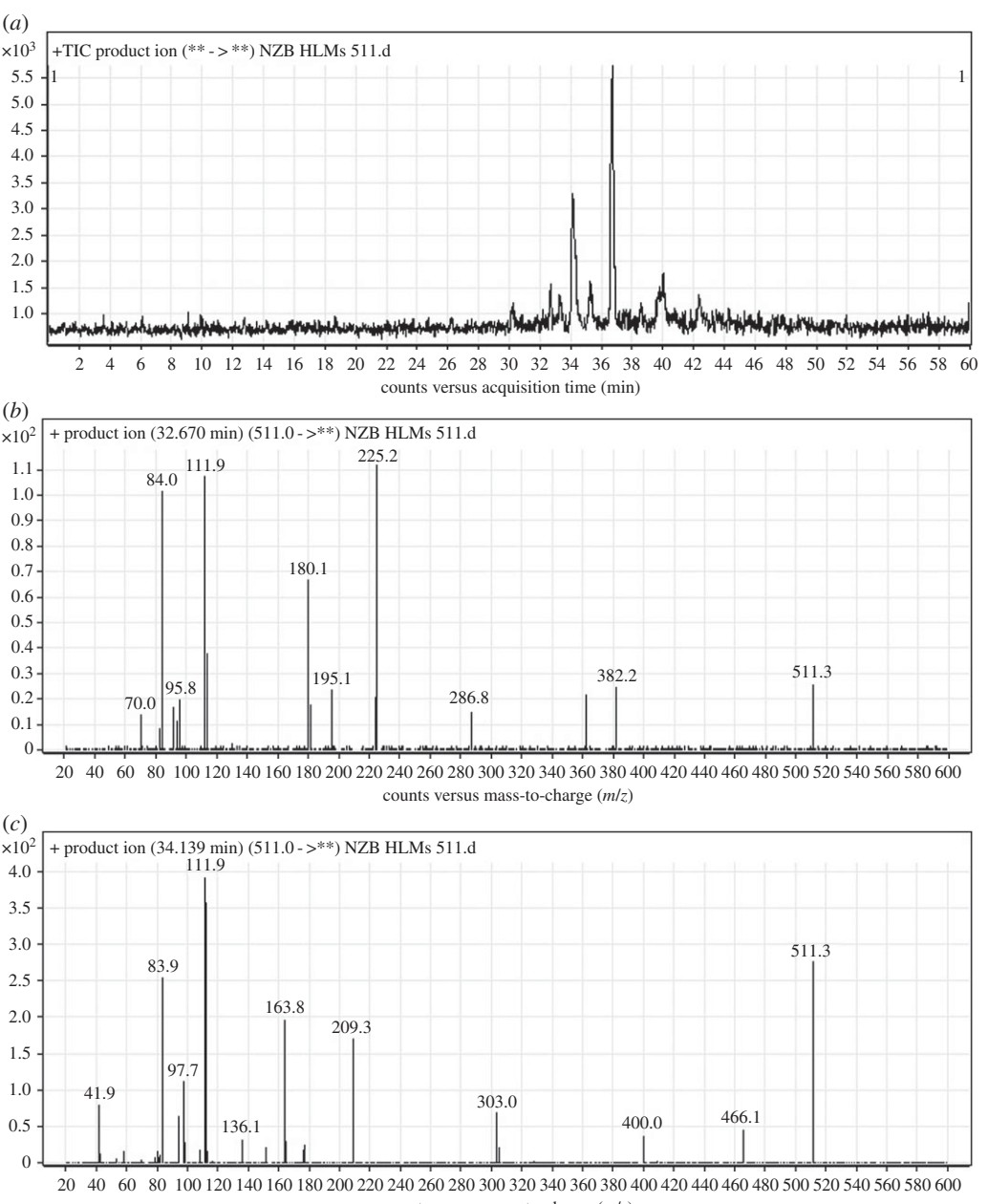

**Figure 5.** DI chromatogram of NZB511 metabolites (*a*) and DI mass spectra at 32.76 min (*b*) and 34.14 min (*c*).

**Scheme 6.** Fragmentation behaviour of NZB511a. DIs, daughter ions.

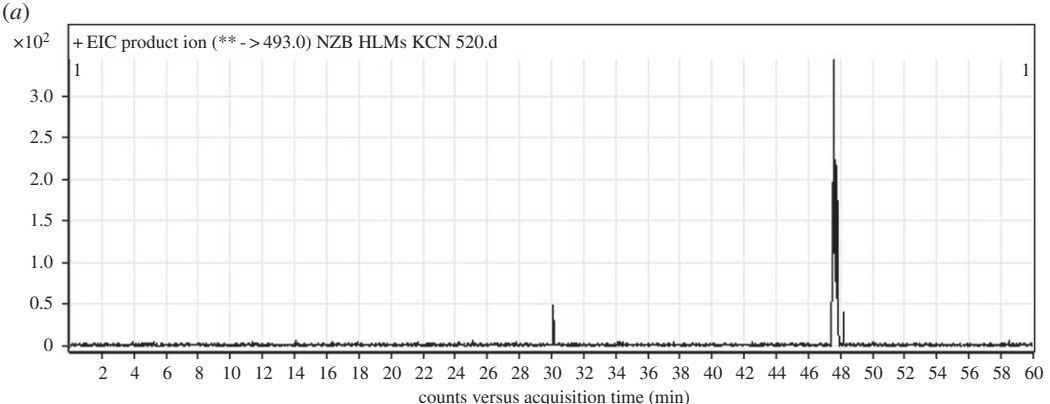

**Scheme 7.** Fragmentation behaviour of NZB511b. DIs, daughter ions.

*(a)*

×10² + EIC product ion (\*\* ->493.0) NZB HLMs KCN 520.d

counts versus acquisition time (min)

*(b)*

×10² + product ion (47.664 min) (520.0 ->\*\*) NZB HLMs KCN 520.d

counts versus mass-to-charge (*m/z*)

**Figure 6.** DI chromatogram of NZB520 (*a*) and DI mass spectrum at 47.66 min (*b*).

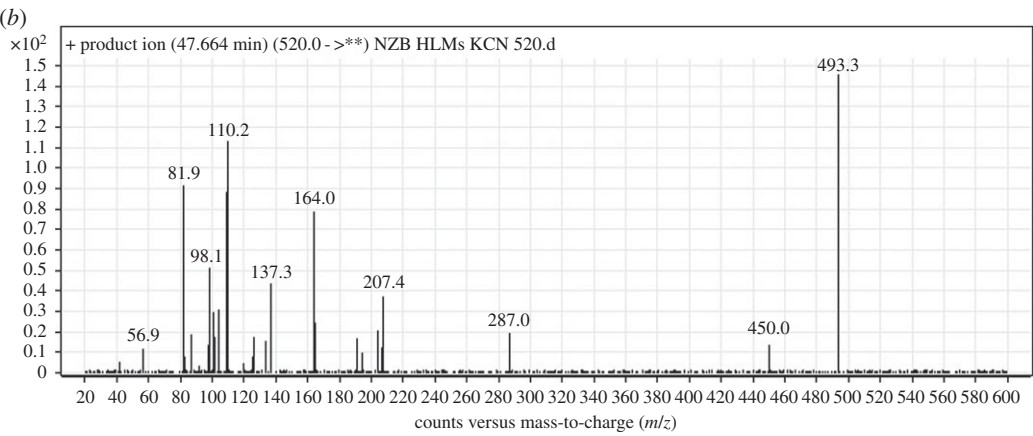

**Scheme 8.** Fragmentation behaviour of NZB520. DIs, daughter ions.

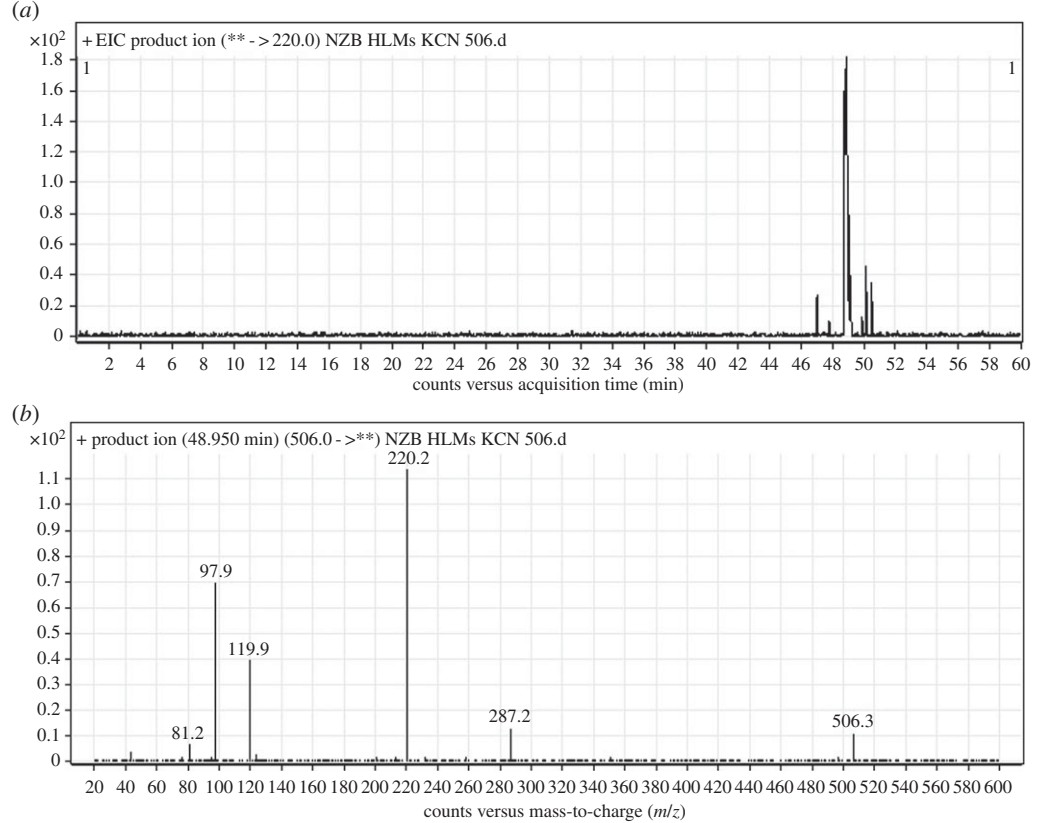

**Figure 7.** DI chromatogram of NZB506 (*a*) and DI mass spectrum at 48.95 min (*b*).

**Scheme 9.** Fragmentation behaviour of NZB506. DIs, daughter ions.

### 3.3.1. Identification of the NZB520 cyano adduct

The NZB520 PI peak eluted at 47.66 min (figure 6*a*). The fragmentation of the PI at *m*/*z* 520 produced five DIs at *m*/*z* 493, *m*/*z* 207, *m*/*z* 164, *m*/*z* 83 and *m*/*z* 57 (figure 6*b*). The DI at *m*/*z* 493 indicated the loss of 27 *m*/*z* units, representing the neutral loss of a hydrogen cyanide molecule. The DI at *m*/*z* 164 revealed the absence of any metabolic reaction on the azepane ring. The DIs at *m*/*z* 137 and *m*/*z* 83 confirmed that cyanide ion addition occurred on the bioactivated carbon α of the terminal tertiary N atom (dimethyl amine) (scheme 8).

### 3.3.2. Identification of the NZB506 cyano adduct

The NZB506 PI peak eluted at 48.95 min (figure 7*a*). The fragmentation of the PI at *m*/*z* 506 generated four DIs at *m*/*z* 287, *m*/*z* 220, *m*/*z* 120 and *m*/*z* 98 (figure 7*b*). The DI at *m*/*z* 287 indicated the absence of any metabolic reaction on the isonicotinamide and benzimidazole groups. The DIs at *m*/*z* 220 and *m*/*z* 98

**Scheme 10.** Proposed pathway of bioactivation during NZB metabolism by human liver microsomes and the cyanide trapping strategy.

confirmed the addition of a cyanide ion on the activated carbon α of the terminal tertiary N atom (dimethyl amine) and N-demethylation of the dimethyl amine group (scheme 9).

## 3.4. Bioactivation mechanism of NZB

The characterization of the NZB506 and NZB520 cyano adducts revealed the generation of reactive iminium intermediates in NZB metabolism. The hydroxylation of the bioactivated carbon in NZB followed by dehydration resulted in the generation of reactive iminium electrophiles that were captured by a cyanide nucleophile to form a stable cyano adduct (scheme 10). The bioactivation pathway for the formation of reactive intermediates has been previously studied using drugs containing cyclic tertiary amines. However, herein, the reactive intermediates were generated by bioactivation of an aliphatic noncyclic carbon attached to a tertiary amine rather than by azepane bioactivation [39–44].

## 4. Conclusion

The current study provided experimental evidence to support further work on NZB toxicity. Six *in vitro* NZB phase I metabolites and two cyano adducts were identified (figure 8) and bioactivation mechanisms

**Figure 8.** Chemical structure of NZB showing the sites of phase I metabolic reactions responsible for the generation of the detected metabolites. The main bioactive centre is indicated by an asterisk.

were proposed. The knowledge on bioactivation mechanisms is crucial for determining the chemical groups involved in bioactivation. This information may be used for the development of new molecules containing small sterically hindering groups, isosteric replacement or a spacer to prevent NZB bioactivation; inhibiting the generation of reactive species in this way would result in reduced toxicity. The data obtained in this study will contribute towards the development of new drugs with enhanced safety profiles.

Ethics. The study's design (*in vitro* assays using commercially available liver microsomes) exempts it from the approval by Ethics Committees.

Data accessibility. The data supporting the results in this article can be accessed at the Dryad Digital Repository: https://doi.org/10.5061/dryad.j5m8h10 [45].

Authors' contributions. A.A.K. and A.S.A. designed and supervised the study. A.S.A., M.W.A. and A.A.K. performed the optimization for the experimental steps and protocol. M.W.A. conducted the experiments and drafted the manuscript. All authors revised and approved the final version of the manuscript. All authors agreed with the submission of the manuscript to Royal Society Open Science Journal.

Competing interests. The authors declare no competing interests.

Funding. The authors thank the Deanship of Scientific Research at King Saud University for funding this work through Research Group Project no. RG-1435-025.

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
