## [Reviewer comments · Royal Society Open Science]

Review History

RSOS-190852.R0 (Original submission)

Review form: Reviewer 1

Is the manuscript scientifically sound in its present form?

Yes

Are the interpretations and conclusions justified by the results?

Yes

Is the language acceptable?

Yes

Do you have any ethical concerns with this paper?

No

Recommendation?

Accept with minor revision (please list in comments)

Comments to the Author(s)

Title: Acceptable, author shouldn't abbreviate liquid chromatography tandem mass spectrometry into LC-MS/MS following journal guidelines. LC-MS/MS should be expanded.

Abstract: Acceptable

Abbreviations should not be used in the abstract. Spare this abbreviation to first appearance in the text. Abbreviations should be provided in a separate section.

Keywords: Acceptable

Remove chemotherapy from keywords.

Introduction:

Separate paragraph should be included explaining the side effects of the formed reactive metabolites with similar drugs in the literature.

Methods:

1- Table 1 should be removed and all chromatographic conditions should be mentioned in a text.

2- Model number of liquid chromatography and mass spectrometry instruments should be mentioned.

3- Author should give more details on type of controls used in incubations.

4- In vitro and m/z should be italic all over the manuscript including text, figures and schemes.

Results:

1- Figures: Acceptable.

2- Schemes: acceptable, font should be unified and m/z should be italic.

Conclusions: It should be concise as it contains duplication from the manuscript sentences.

References: Journal names should be abbreviated.

Review form: Reviewer 2

Is the manuscript scientifically sound in its present form?

Yes

Are the interpretations and conclusions justified by the results?

Yes

Is the language acceptable?

Yes

Do you have any ethical concerns with this paper?

No

Recommendation?

Accept as is

Comments to the Author(s)

It is well designed and clear paper. In my opinion, it is acceptable.

Review form: Reviewer 3

Is the manuscript scientifically sound in its present form?

Yes

Are the interpretations and conclusions justified by the results?

Yes

Is the language acceptable?

Yes

Do you have any ethical concerns with this paper?

Yes

Recommendation?

Accept with minor revision (please list in comments)

Comments to the Author(s)

1. This paper was introduced the six NZB phase I metabolites, formed by hydroxylation, oxidation, and N-demethylation, were characterized. Using the LC-MS/MS metabolic profiling of nazartinib, a novel anticancer drug, reveals the formation of unexpected reactive metabolites. All of these Chemical compounds were qualitatively by LC-MS/MS. At the beginning, Do you make some attempts to the test these chemical compounds by the GC-MS/MS? Because the GC-MS/MS is better than LC-MS/MS in determine the chemical composition of a substance. By now We can not search for relevant papers about the determination of the NZB and the relevant metabolites. The EI and ESI were different, this result in the GC-MS/MS has the standard ion library.

2. The Keywords. I think you had better add the LC-MS/MS.

3. The table 1. I suggest you to search for some analytical chemistry papers. Most of these were written in the "Material and methods" for a Instrumentation by scripts.

4. The table 1. Gradient steps should be Gradients.

5. This paper "P2.03-012 Characterization of the Efficacies of Osimertinib and Nazartinib against Cells Expressing Epidermal Growth Factor Receptor Mutations. " was early introduced the Nazartinib against the Epidermal Mutations. Journal of Thoracic Oncology. K. Masuzawa H. Yasuda J. Hamamoto et al. This may add into the introduction reference.

In general, These experiments have not been done before. But this paper should be carefully revised.

Decision letter (RSOS-190852.R0)

01-Jul-2019

Dear Dr Attwa:

Title: LC-MS/MS metabolic profiling of nazartinib, a novel anticancer drug, reveals the formation of unexpected reactive metabolites

Manuscript ID: RSOS-190852

Thank you for submitting the above manuscript to Royal Society Open Science. On behalf of the Editors and the Royal Society of Chemistry, I am pleased to inform you that your manuscript will

be accepted for publication in Royal Society Open Science subject to minor revision in accordance with the referee suggestions. Please find the reviewers' comments at the end of this email.

The reviewers and handling editors have recommended publication, but also suggest some minor revisions to your manuscript. Therefore, I invite you to respond to the comments and revise your manuscript.

Because the schedule for publication is very tight, it is a condition of publication that you submit the revised version of your manuscript before 10-Jul-2019. Please note that the revision deadline will expire at 00.00am on this date. If you do not think you will be able to meet this date please let me know immediately.

Best wishes,
Dr Laura Smith
Publishing Editor, Journals

RSC Associate Editor:
Comments to the Author:
(There are no comments.)

RSC Subject Editor:
Comments to the Author:
(There are no comments.)

Reviewer comments to Author:
Reviewer: 1

Comments to the Author(s)

Title: Acceptable, author shouldn't abbreviate liquid chromatography tandem mass spectrometry into LC-MS/MS following journal guidelines. LC-MS/MS should be expanded.

Abstract: Acceptable

Abbreviations should not be used in the abstract. Spare this abbreviation to first appearance in the text. Abbreviations should be provided in a separate section.

Keywords: Acceptable
Remove chemotherapy from keywords.

Introduction:

Separate paragraph should be included explaining the side effects of the formed reactive metabolites with similar drugs in the literature.

Methods:

- 1- Table 1 should be removed and all chromatographic conditions should be mentioned in a text.
- 2- Model number of liquid chromatography and mass spectrometry instruments should be mentioned.
- 3- Author should give more details on type of controls used in incubations.
- 4- In vitro and *m/z* should be italic all over the manuscript including text, figures and schemes.

Results:

- 1- Figures: Acceptable.
- 2- Schemes: acceptable, font should be unified and *m/z* should be italic.

Conclusions: It should be concise as it contains duplication from the manuscript sentences.
References: Journal names should be abbreviated.

Reviewer: 2

Comments to the Author(s)

It is well designed and clear paper. In my opinion, it is acceptable.

Reviewer: 3

Comments to the Author(s)

1. This paper was introduced the six NZB phase I metabolites, formed by hydroxylation, oxidation, and N-demethylation, were characterized. Using the LC-MS/MS metabolic profiling of nazartinib, a novel anticancer drug, reveals the formation of unexpected reactive metabolites. All of these Chemical compounds were qualitatively by LC-MS/MS. At the beginning, Do you make some attempts to the test these chemical compounds by the GC-MS/MS? Because the GC-MS/MS is better than LC-MS/MS in determine the chemical composition of a substance. By now We can not search for relevant papers about the determination of the NZB and the relevant metabolites. The EI and ESI were different, this result in the GC-MS/MS has the standard ion library.

2. The Keywords. I think you had better add the LC-MS/MS.

3. The table 1. I suggest you to search for some analytical chemistry papers. Most of these were written in the "Material and methods" for a Instrumentation by scripts.

4. The table 1. Gradient steps should be Gradients.

5. This paper "P2.03-012 Characterization of the Efficacies of Osimertinib and Nazartinib against Cells Expressing Epidermal Growth Factor Receptor Mutations." was early introduced the Nazartinib against the Epidermal Mutations. Journal of Thoracic Oncology. K. Masuzawa H. Yasuda J. Hamamoto et al. This may add into the introduction reference.

In general, These experiments have not been done before. But this paper should be carefully revised.

Author's Response to Decision Letter for (RSOS-190852.R0)

See Appendix A.

Decision letter (RSOS-190852.R1)

22-Jul-2019

Dear Dr Attwa:

Title: Liquid chromatography-tandem mass spectrometry metabolic profiling of nazartinib reveals the formation of unexpected reactive metabolites

Manuscript ID: RSOS-190852.R1

It is a pleasure to accept your manuscript in its current form for publication in Royal Society

Open Science. The chemistry content of Royal Society Open Science is published in collaboration with the Royal Society of Chemistry.

RSC Associate Editor
Comments to the Author:
(There are no comments.)

Reviewer(s)' Comments to Author:

Appendix A

[6th July 2019]

[Dr Laura Smith]

[Publishing Editor, Journals]

[Royal Society of Chemistry Journal]

We wish to re-submit the revised version of the manuscript titled “**LC-MS/MS metabolic profiling of nazartinib, a novel anticancer drug, reveals the formation of unexpected reactive metabolites.**” The manuscript ID is RSOS-190852.

We thank the reviewers for their thoughtful suggestions and valuable comments regarding our work. The manuscript has benefited from these insightful suggestions. I look forward to working with you and the reviewers to move this manuscript closer to publication in *Royal Society of Chemistry Journal*.

The manuscript has been rechecked and the necessary changes have been made in accordance with the reviewers’ suggestions. The point-by-point responses to the comments raised by the respective reviewers have been prepared and attached below. All changes made are highlighted in the revised manuscript.

Thank you for your consideration. I look forward to hearing from you.

Sincerely,

Mohamed W. Attwa

Department of Pharmaceutical Chemistry, College of Pharmacy, King Saud University, P.O. Box 2457 Riyadh, 11451, Saudi Arabia

Tel.: +966 1146 70237

Fax: +966 1146 76 220

E-mail: mzeidan@ksu.edu.sa

REVIEWER REPORT(S):

Reviewer 1

Dear reviewer:

We thank the reviewer for these pertinent questions and have addressed them all in the revised manuscript.

Comments to the Author(s)

Title: Acceptable, author shouldn't abbreviate liquid chromatography tandem mass spectrometry into LC-MS/MS following journal guidelines. LC-MS/MS should be expanded.

We did as requested. We expanded LC-MS/MS into liquid chromatography tandem mass spectrometry.

Abstract: Acceptable

Abbreviations should not be used in the abstract. Spare this abbreviation to first appearance in the text. Abbreviations should be provided in a separate section.

We did as requested. We added a separate section for abbreviation.

Keywords: Acceptable

Remove chemotherapy from keywords.

We did as requested.

Introduction:

Separate paragraph should be included explaining the side effects of the formed reactive metabolites with similar drugs in the literature.

We thank the reviewer for this recommendation. We added more information regarding side effects of the formed reactive metabolites.

Methods:

1- Table 1 should be removed and all chromatographic conditions should be mentioned in a text.

Table 1 was removed and we added a new section.

2- Model number of liquid chromatography and mass spectrometry instruments should be mentioned.

We thank the reviewer for this recommendation. We did as requested.

3- Author should give more details on type of controls used in incubations.

We did as requested. Controls were prepared following the same steps except the addition of the drug or NADPH.

4- In vitro and m/z should be italic all over the manuscript including text, figures and schemes.

We did as requested.

Results:

1- Figures: Acceptable.

2- Schemes: acceptable, font should be unified and m/z should be italic.

We did as requested.

Conclusions: It should be concise as it contains duplication from the manuscript sentences.

We rephrase the conclusion to be more concise.

References: Journal names should be abbreviated.

We did as requested. All journal names were abbreviated.

Reviewer 2

Dear reviewer:

We thank the reviewer for the acceptance of our manuscript.

Comments to the Author(s)

It is well designed and clear paper. In my opinion, it is acceptable.

Reviewer 3

Dear reviewer:

We thank the reviewer for these pertinent questions and have addressed them all in the revised manuscript.

Comments to the Author(s)

1. This paper was introduced the six NZB phase I metabolites, formed by hydroxylation, oxidation, and N-demethylation, were characterized. Using the LC-MS/MS metabolic profiling of nazartinib, a novel anticancer drug, reveals the formation of unexpected reactive metabolites. All of these Chemical compounds were qualitatively by LC-MS/MS. At the beginning, Do you make some attempts to the test these chemical compounds by the GC-MS/MS? Because the GC-MS/MS is better than LC-MS/MS in determine the chemical composition of a substance. By now We cannot search for relevant papers about the determination of the NZB and the relevant metabolites. The EI and ESI were different, this result in the GC-MS/MS has the standard ion library.

We thank the reviewer for this comment.

Regarding GC-MS/MS:

1- We don't have GC-MS/MS in our laboratory.

2- There is no library for unknown metabolites with unknown chemical structures.

3- LC-MS/MS is considered the standard technique for identification of unknown metabolites.

2.The Keywords. I think you had better add the LC-MS/MS.

We added LC-MS/MS to keywords as requested.

3.The table 1. I suggest you to search for some analytical chemistry papers. Most of these were written in the “Material and methods” for a Instrumentation by scripts.

We thank the reviewer for this recommendation. Table 1 was removed and replaced by chromatographic separation section.

4.The table 1. Gradient steps should be Gradients.

The error was corrected as requested.

5.This paper “P2.03-012 Characterization of the Efficacies of Osimertinib and Nazartinib against Cells Expressing Epidermal Growth Factor Receptor Mutations. “ was early introduced the Nazartinib against the Epidermal Mutations. Journal of Thoracic Oncology.K.Masuzawa H.Yasuda J.Hamamoto et al. This may add into the introduction reference.

We updated this reference in the introduction.